# Transcriptomic Analysis Provides Insights into Anthocyanin Accumulation in Mulberry Fruits

**Rongli Mo** [1,†], **Na Zhang** [1,†], **Jinxin Li** [1], **Qiang Jin** [2] 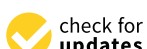, **Zhixian Zhu** [1], **Zhaoxia Dong** [1], **Yong Li** [1], **Cheng Zhang** [1] **and Cui Yu** [1,*]

1    Institute of Economic Crops, Hubei Academy of Agricultural Sciences, Wuhan 430064, China
2    College of Horticulture and Forestry, Tarim University, Alar 843300, China
\*    Correspondence: mrsyu888@hotmail.com; Tel.: +86-27-8710-6003
†    These authors contributed equally to this work.

**Abstract:** Mulberry fruits are rich in anthocyanins, which are important secondary metabolites that give mulberries their bright color, favorable taste and high nutritional quality, making them a popular fruit. However, few studies have focused on the molecular mechanism underlying anthocyanin accumulation in mulberries and the gene regulatory networks of anthocyanin biosynthetic pathways remain largely unknown. In this study, we performed RNA sequencing to identify differentially expressed genes (DEGs) associated with anthocyanin accumulation between two mulberry genotypes ('Zi Jing', ZJ and 'Zhen Zhu Bai', ZZB, with purple and white fruit flesh, respectively) at 5, 18, 27 and 31 days after flower. Using transcriptome analysis, we explored several key DEGs involved in the anthocyanin biosynthetic pathway, including the structural genes: *CHS*, *CHI*, *F3H*, *DFR1*, *DFR2* and *ANS*, known as MBW complex genes: *MYB* (*M.alba_G0017209*), *MYB* (*M.alba_G0017689*), *bHLH* (*M.alba_G0012659*), *bHLH* (*M.alba_G0009347*) and *bHLH3* (*M.alba_G0016257*) and the ethylene response factor: *ERF* (*M.alba_G0016603*). Of these, changing trends related to expression pattern and anthocyanin content showed their most positive correlation at the post-flowering stage in both genotypes. Our results indicated that ethylene enhances anthocyanin accumulation in mulberry fruits. Furthermore, qRT-PCR was performed to confirm the above-mentioned genes' expression (except for *MYB* (*M.alba_G0017689*) and *bHLH* (*M.alba_G0009347*) was significantly up-regulated under ethylene treatment at 300 mg/L. These findings help uncover the gene regulatory networks of the anthocyanin biosynthetic pathway and will contribute to engineering purposes in future mulberry breeding programs.

**Keywords:** mulberry fruit; anthocyanins biosynthetic pathway; RAN-seq; ethylene; MYB; bHLH; ERF

## 1. Introduction

Mulberry (*Morus* L.) belongs to the Moraceae family. Its leaves are used as the main food for domesticated silkworm (*Bombyx mori* L.) [1] and its fruits have been used as a medicinal food to improve human health in China due to their taste and high nutritional value [2,3]. Anthocyanins are the most abundant nutrient components in purple mulberry fruits [4] and they play equally important roles in protecting human health due to their anti-inflammatory, anti-cancer and antioxidant properties, as well as their role in lowering blood pressure and improving vision [5].

As water-soluble secondary metabolites in plants, anthocyanins are localized in vesicles and are the most important pigments in fruits and flowers, giving plant tissues a red or blue coloring [6]. The anthocyanin synthesis process, which occurs via the phenylpropanoid pathway, has been elucidated [7–10] and includes early flavonoid biosynthesis pathway genes (*PAL*, *CHS* and *F3'H*) and late biosynthetic genes (*DFR*, *ANS* and *UFGT*). The synthesis-involved genes *ANS*, *F3'H1*, *F3H1*, *CHI* and *CHS1* are correlated with anthocyanin concentrations during mulberry fruit ripening [4,11]. The MYB-bHLH-WD40

(MBW) transcriptional complex is the primary regulator of anthocyanin biosynthesis [12]. It has been reported that bHLH3 is a key positive regulator for mulberry fruit color and that MBW-activated MYB4 is involved in the negative feedback control of the regulatory network, balancing the accumulation of anthocyanins and proanthocyanidins [13].

However, it is unclear how ethylene is associated with the coloration of purple mulberry fruit during ripening. Phytohormones are also involved in the biosynthesis of anthocyanins [9]. Ethylene is an important regulating factor during fruit ripening, especially by controlling the reduction in chlorophyll and the accumulation of anthocyanins or carotenoids [14–16]. The ethylene response factor (ERF) acts at the end of the ethylene biosynthesis pathway, regulating pigment synthesis, fruit softening, flavor and aroma formation during fruit ripening [17–20]. MdERF3 works as a key factor regulating ethylene synthesis in apples by participating in anthocyanin biosynthesis under the transcriptional regulation of MdMYB1 [21]. *ERF* genes from pears, such as *PyERF3*, *Pp4ERF24* and *Pp12ERF96*, regulate anthocyanin accumulation via interactions with *MYB114* and *bHLH3* during fruit ripening [22,23]. However, it remains unclear how ethylene is associated with the coloration of purple mulberry fruit during ripening.

In this study, we used transcriptome profiling to elucidate the dynamics of fruit color transitions in two mulberry genotypes with contrasting fruit colors. Global gene expression analysis was performed for the successful identification of the key structural and regulatory genes involved in anthocyanin biosynthesis during the development of mulberry fruit development. The role of plant hormone ethylene inducing the anthocyanins accumulation in mulberry fruits was verified by ethylene treatment in vivo and in vitro. Furthermore, the profiling of the differentially expressed *ERF* gene family between two mulberry genotypes during different fruit coloring stages was surveyed. Combined with correlation analysis of transcription expression abundance and anthocyanin content, the key *ERF* gene involved in anthocyanin accumulation was further validated by ethylene (300 mg/L)-treated fruit samples in vitro. This provides a theoretical basis to help better understand the molecular mechanisms of anthocyanin accumulation during fruit ripening in mulberry.

## 2. Materials and Methods

### 2.1. Plant Material

Two varieties of Morus L., purple mulberry cultivar 'Zi Jing' (ZJ) (*Morus multicaulis* P.) and the white mulberry cultivar 'Zhen Zhu Bai' (ZZB) (*Muros alba* L.) were used in this study and were obtained from a mulberry germplasm resource nursery in Industrial Crops Institute of Hubei Academy of Agricultural Sciences, Wuhan, China. Two cultivar fruits were harvested at 5, 10, 18, 27 and 31 days after flower (DAF), respectively, to measure the anthocyanin content, ethylene content, fruit solidity and soluble solids content. Furthermore, the fruit samples at 5, 18, 27 and 31 DAF between ZJ and ZZB were collected for transcriptome sequencing analysis. However, the fruits of the purple mulberry cultivar ZJ at 10 DAF in vivo and in vitro were sampled for ethylene treatment with 0, 100, 300 and 500 mg/L, respectively. The control group was treated with $ddH_2O_2$. Finally, the treated fruits were incubated in vitro at 25 °C for 0, 48, 72 and 84 h and were allowed to grow in vivo under ambient temperatures (20–28 °C) for 0, 48 and 72 h, respectively. Each sample was composed of three biological replicates and at least 20 fruits were sampled for each biological replicate.

### 2.2. Measurement of Physiological Indicators

Measuring the anthocyanin content was performed according to a previously described method [24], with some modifications. Anthocyanins were isolated from 2 mL of 1% HCL–methanol solution with 0.5 g mulberry fruit at 4 °C under dark conditions [25].

Ethylene content, fruit firmness and soluble solids content (SSC) were examined as previously described [26]. To evaluate the ethylene-production rate, four sealed 50 mL bottles with 20 g of fruits in each were maintained at 25 °C for 4 h and then 1 mL of headspace gas was obtained with an airtight syringe to measure the ethylene content using

a GC2010 gas chromatograph (Shimadzu, Kyoto, Japan). Regarding fruit firmness, two mulberry genotypes were examined by a CT3 texture analyzer (Brookfield, 3375 North Delaware Street, Chandler, AZ, USA), in which the probe was inserted into a 3 mm depth of ten fruits to test the fruit's ripeness. Subsequently, 200 mL pressed fruit samples was obtained to check the SSC using a PAL-1 refractometer (Atago, Tokyo, Japan).

### 2.3. Construction and Sequencing of Illumina RNA-seq Library

The RNA extraction was performed using TRIzol according to the manufacturer's instructions (Invitrogen, Carlsbad, CA, USA), in which DNA was removed with DNase I (TaKara, Forster City, CA, USA) as described [27]. RNA Libraries were constructed with 24 samples from four developmental stages (at 5, 18, 27 and 31 DAF) of two mulberry genotypes according to the procedures of Wuhan Frasergen Bioinformatics Co., Ltd. (China). RNA-seq was analyzed on an Illumina Hiseq X Ten platform.

### 2.4. Data Analysis of RNA-Seq

As described in our published literature [28], adapters and low-quality reads (35 bp) were first discarded using the software Trim-galore v0.6.2 (https://github.com/FelixKrueger/Trim-100 Galore, accessed on 15 June 2022). Then, the quality reads were merged after removing repeats and the high-quality reads were assembled using Trinity (version r20140717, http://www.Trinityrnaseq.github.io/, accessed on 15 June 2022) to create a specific transcript. Differentially expressed genes between the two genotypes were analyzed using DESeq (http://www.bioconductor.org/, accessed on 20 June 2022) according to previous procedures [29], in which DEGs were considered to be differentially expressed genes with $p < 0.01$ and FC > 2. Fragments Per Kilobase of exon model per million mapped fragments was used to normalize the transcript level.

### 2.5. KEGG, GO and WGCNA Analysis

The obtained DEGs were analyzed and annotated in Nr, Nt and SWISS-PROT databases using previous procedures [28]. Protein domains were annotated using the ortholog groups clusters of proteins database (COG; E-values $1 \times 10^{-10}$, using rpsBlast), (KEGG, release 58; E-values $1 \times 10^{-10}$). Protein domains were annotated by InterProScan Release 36.0 annotated protein domains and their functional assignments were mapped onto Gene Ontology (GO, http://www.geneontology.org/, accessed on 20 July 2022, using the BlastX algorithm).

Based on the WGCNA analysis, the co-expression networks highly related to expression patterns were established and the transcripts of differentially expressed genes were enriched. The co-expression modules were established using the one-step network construction with default settings following the tutorial [30].

### 2.6. qRT-PCR

Total RNA was isolated from the mulberry fruits of wild type and 10-DAF cultivar ZJ treated with 300 mg/L ethylene. The first-strand cDNA was obtained using a FastKing RT Kit (TIANGEN, Beijing, China) with high-integrity RNA as a template. qRT-PCR was performed using an SYBR Green Master mix on a Light Cycle 96 Real-Time PCR system (Roche, Basel, Switzerland). The relative gene expression levels were calculated by $2^{-\Delta\Delta Ct}$. Table S1 displays the primer sequences used in this study.

## 3. Results

### 3.1. Morphological Profiles and Physiological Characters of the Mulberry Fruit Samples

The fruit color changes of two mulberry varieties, i.e., ZJ (purple color) and ZZB (white color), showed obvious differences in the development of fruits (Figure 1A). As the fruit matured, the ZZB cultivar fruit turned a jade-white color due to the continuous degradation of chlorophyll. However, the difference is that the ZJ fruit continuously accumulated anthocyanins and became a purple-red color. As previously reported, the results suggest that the mulberry fruits are climacteric, showing a gaseous ethylene surge during ripening

(at 27 DAF) (Figure 1B). Therefore, with a large accumulation of ethylene, the fruit firmness of the two varieties rapidly decreased from 18 to 31 DAF (Figure 1C). Additionally, an increase in the SSC of both mulberry cultivars (Figure 1D) and the anthocyanin content from ZJ fruits was observed (Figure 1E) after the ethylene respiration peak.

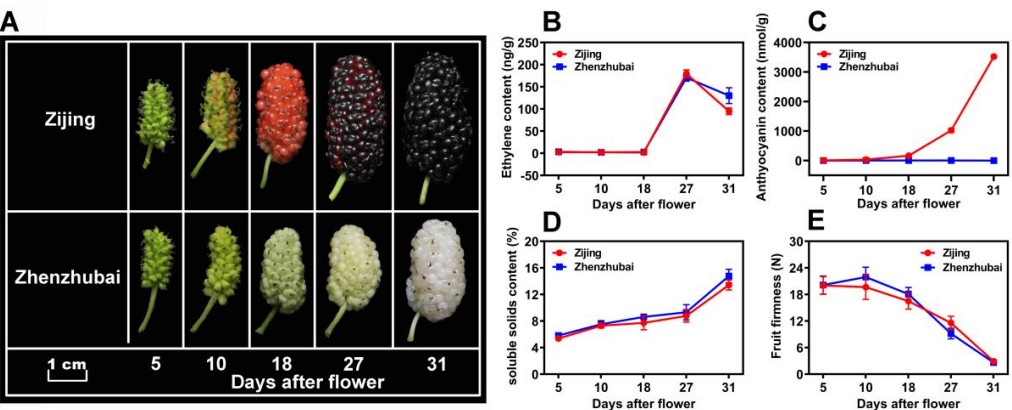

**Figure 1.** Fruit morphologies and physiological trends of Mulberry 'Zi Jing' (ZJ) and 'Zhen Zhu Bai' (ZZB) at different developmental stages. (**A**) Morphological profiles of the mulberry fruits from ZZB and ZJ. (**B–E**) Ethylene content, anthocyanin content, soluble solids content and fruit firmness, respectively, for ZJ and ZZB at different DAF. The white horizontal bar in panel A represents 1 cm in length. Each data point of the curves in panels (**B–E**) represents the mean of three independent replicates (±SE).

### 3.2. Data Analysis of RNA Sequencing

To better learn the expression pattern of anthocyanin biosynthesis genes in ZJ and ZZB fruits, RNA sequencing was performed with fruits from 5, 18, 27 and 31 DAF, respectively (three biological replicates for each sample), and a total of 136.61 GB of clean data was obtained (high quality of sequencing, as shown in Table S2). The mapped rate between these clean data and the *Muros alba* L. reference genome ranged from 89.27 to 91.8%, with an average of 90.55% (Table S2), indicating no pollution occurred during experiments.

To gain more insights about the DEGs and taking into account the developmental factors between two mulberry genotypes, we compared the DEGs for each genotype at the last three post-flowering stages (18, 27 and 31 DAF) relative to stage I (5 DAF) (Figure 2B). Notably, 3375 and 4703 overlapped genes were identified over the three post-flowering stages for ZJ and ZZB, respectively (Figure 2A). Furthermore, we compared the DEGs for two genotypes at four post-flowering stages (5, 18, 27 and 31 DAF) (Figure 2A). Notably, 270 overlapped genes were identified over the four post-flowering stages between both ZJ and ZZB and 1048, 1241, 805 and 788 unique differentially expressed genes were observed between four comparison groups of ZZB-5 vs. ZJ-5, ZZB-18 vs. ZJ-18, ZZB-27 vs. ZJ-27 and ZZB-31 vs. ZJ-31, respectively (Figure 2A). The lowest number of DEGs was obtained between the ZJ-5 and ZZB-5 samples, while the highest numbers of DEGs were detected between the ZZB-5 and ZZB-31 samples (Figure 2B). Circos plots show the location of these DEGs in chromosomes (Figure 2C) and heatmaps show the gene expression levels from ZZB and ZJ at 5, 18, 27 and 31 DAF, respectively (Figure 2D).

The WGCNA was performed to reveal the interconnected gene sets that were associated with anthocyanin accumulation. Transcripts were grouped into thirteen co-expression modules (Figure 3A). The red module, pink module and tan module exhibited high expression levels at the last two post-flowering stages (27 and 31 DAF) in the ZJ genotype, especially the red module (Figure 3B). Pearson correlation analysis indicated that the red module showed strong correlation with the anthocyanin dynamic content, PCC (*r*) was 0.86 (Figure 3C). However, a KEGG analysis performed on the red module showed that flavonoid biosynthesis was significantly enriched (Figure 3D).

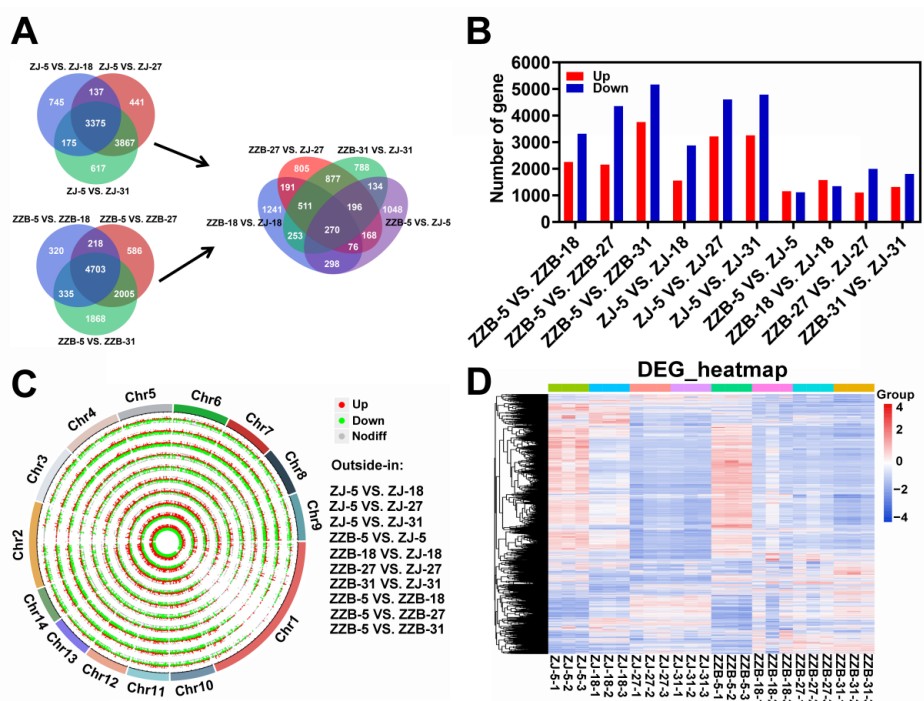

**Figure 2.** Identification and analysis of DEGs between ten comparison groups of ZZB-5 vs. ZZB-18, ZZB-5 vs. ZZB-27, ZZB-5 vs. ZZB-31, ZJ-5 vs. ZJ-18, ZJ-5 vs. ZJ-27, ZJ-5 vs. ZJ-31, ZZB-5 vs. ZJ-5, ZZB-18 vs. ZJ-18, ZZB-27 vs. ZJ-27 and ZZB-31 vs. ZJ-31 of ZJ and ZZB at different DAF stages, respectively. (**A**) Venn diagram representing the overlapped genes and unique genes. (**B**) The gene numbers were up- and down-regulated. (**C**) Genome-wide distribution of DEGs on a chromosomal scale. (**D**) Heatmap of DEGs from ZZB and ZJ at 5, 18, 27 and 31 DAF, respectively.

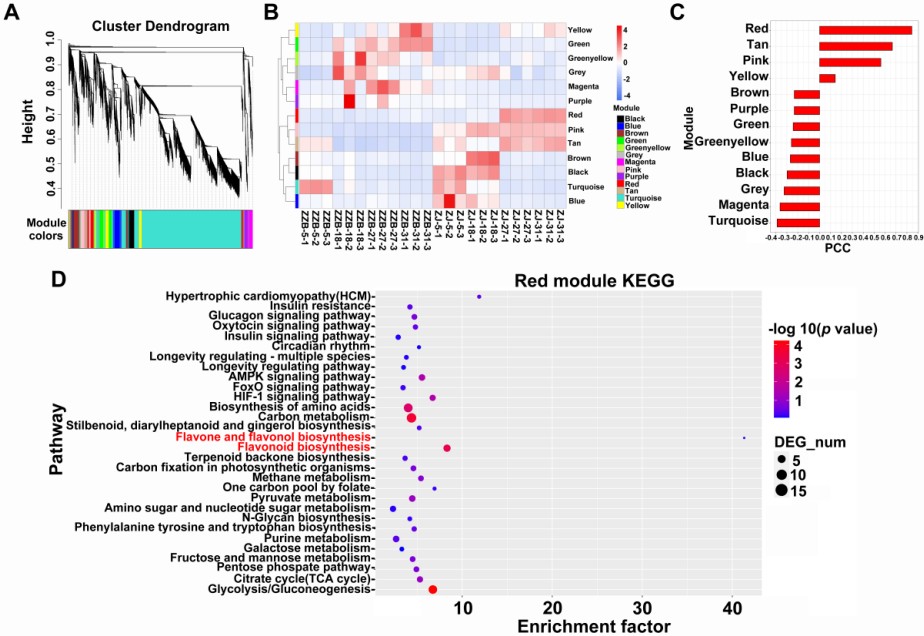

**Figure 3.** WGCNA analysis of transcripts in ZJ and ZZB genotypes during fruit development. (**A**) Hierarchical clustering tree. (**B**) Module eigengene expression. (**C**) Pearson correlation coefficient, PCC (*r*) between the red module and anthocyanin content in ZJ and ZZB at different post-flowering stages. (**D**) KEGG enrichment analysis of the red module.

### 3.3. Identification and Enrichment Analysis of DEGs between the Two Mulberry Genotypes at 27 DAF

To identify the genes closely related to anthocyanin biosynthesis, we selected ZZB-27 and ZJ-27 fruit samples, where the anthocyanin content was sharply increased in ZJ (Figure 1C) for DEG analysis. To gain insight into the transcriptome of anthocyanin biosynthesis, we performed Pearson correlation analysis (Figure 4A) and the Pearson correlation coefficients were all above 0.8 between the three biological replicates of samples from ZJ and ZZB at 27 DAF (Figure 4A). This indicates that the three biological replicates have very high repeatability. A volcano plot analysis showed that a total of 3104 significant DEGs was identified between the ZJ and ZZB samples at 27 DAF, of which 1107 were upregulated and 1997 were downregulated (Figure 4B). Circos plots show the location of these DEGs in chromosomes (Figure 4C) and heatmaps show the gene expression levels (Figure 4D).

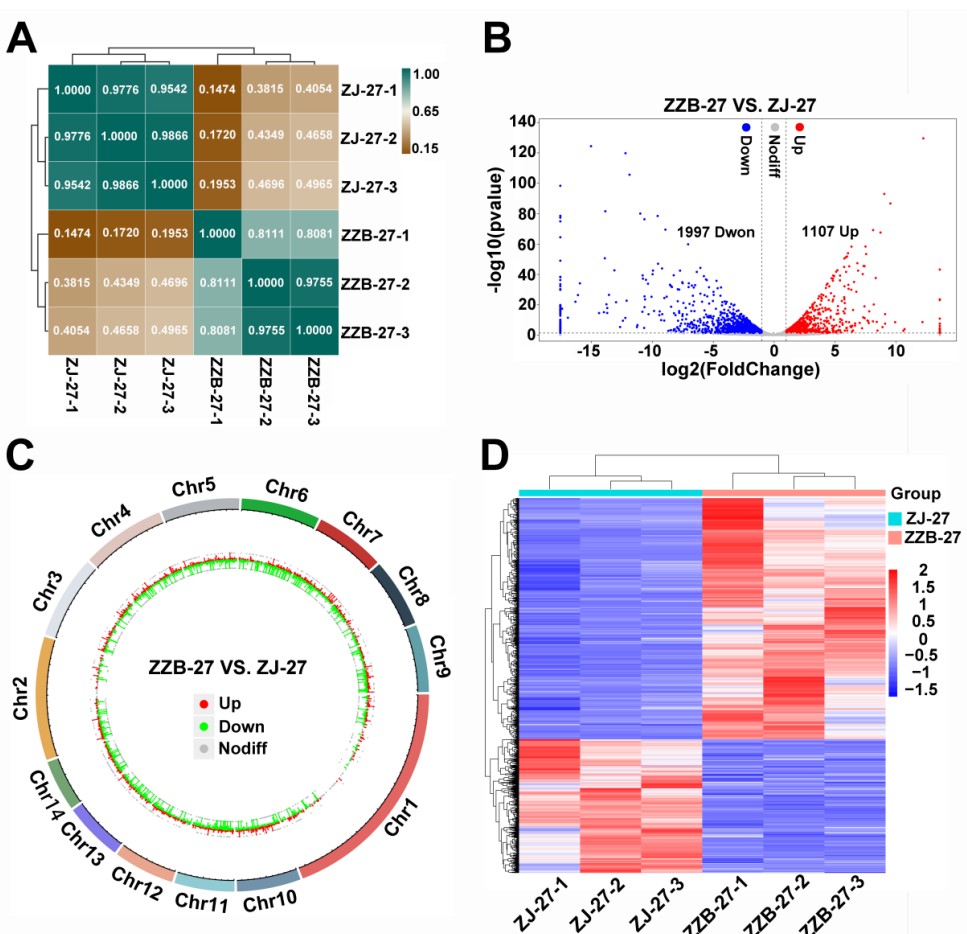

**Figure 4.** Identification and enrichment analysis of DEGs between ZZB and ZJ at 27 DAF. (**A**) Heatmap of samples from ZZB and ZJ with Pearson correlation efficiency at 27 DAF. (**B**) Volcano plot showing that DEGs between ZZB and ZJ at 27 DAF were categorized as up-regulated (red dots), down-regulated (blue dots) or not differentially expressed (gray dots). (**C**) Genome-wide distribution of DEGs on a chromosomal scale. (**D**) Heatmap of DEGs from ZZB and ZJ at 27 DAF.

To clarify the DEG functions, GO and KEGG enrichment was performed to better understand the potential biological pathways targeting fruit phenotypic and physiological differences between ZZB and ZJ (Figure 5). GO types were categorized to be biological processes, cellular components and molecular functions (Figure 5A). When categorized for biological processes, molecular functions and cellular components, the most enriched terms were 'chloroplast grapheme', 'inorganic cation transmembrane transporter activity' and

'plastid translation', respectively. In addition, flavonoid biosynthesis and phytohormone signaling-related genes were significantly enriched during the Top-20 enrichment GO analysis (Figure 5A).

The comparative study of ZZB and ZJ at 27 DAF, combined with the analysis of the top 20 KEGG terms of DEGs, confirmed the presence of α-linolenic acid metabolism, flavonoid biosynthesis, anthocyanin biosynthesis and phytohormone signaling pathways. "Phytohormone signaling" for environmental information processing and "endoplasmic reticulum protein processing" for genetic information processing were the most enriched terms (Figure 5B). In addition, "α-linolenic acid metabolism" and "flavonoid biosynthesis" were two highly enriched terms for metabolism pathways (Figure 5B).

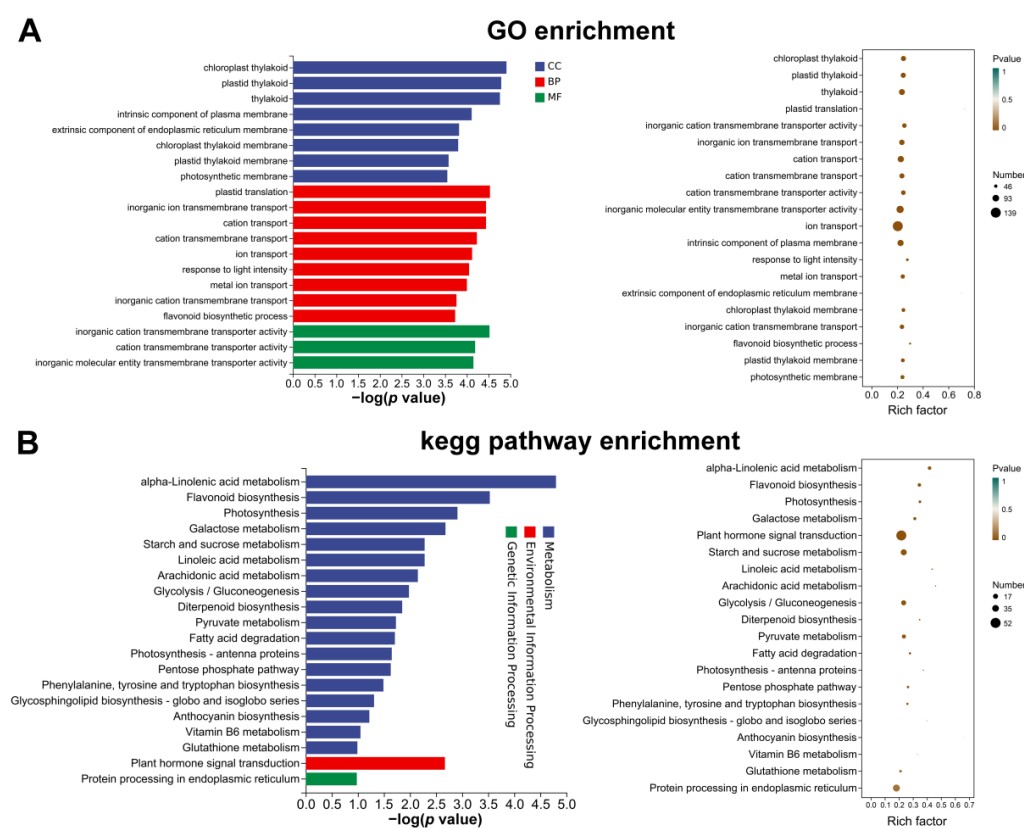

**Figure 5.** GO (**A**) and KEGG (**B**) results of the top-20 enriched genes monitoring the anthocyanin abundance in the fruit of ZJ at 27 DAF compared to ZZB.

### *3.4. Analysis of DEGs Participating in Both Anthocyanin Synthesis Pathway Cultivars*

To further understand the biosynthetic pathway of anthocyanins in Mulberry, 29 DEGs between ZZB and ZJ were identified (Figure S1). Using heatmap analysis, we found that the expression patterns of six enzyme genes were similar to the changes in anthocyanin content as the fruit developed between both cultivars (Figure 6). All the genes directly participate in the biosynthesis process of the anthocyanin biosynthesis, such as chalcone synthase (*CHS*, *M.alba_G0019389*), chalcone isomerase (*CHI, M.alba_G0003811*), flavanone 3-hydroxylase (*F3H, M.alba_G0005697*), dihydroflavonol-4-reductase (*DFR1, M.alba_G0013172; DFR1, M.alba_G0013173*) and anthocyanidin synthase (*ANS, M.alba_G0005420*), all of which were more highly expressed in the ZJ fruit (purple color) as compared to the ZZB fruit (white color) (Figure 6).

Transcription factors (TF) are special representative genes that regulate spatiotemporal expression patterns. The pathway of anthocyanin biosynthesis is mainly controlled by MBW complexes. Using heatmap analysis, we found that five TF genes, such as *MYB* (*M.alba_G0017209*), *MYB* (*M.alba_G0017689*), *bHLH* (*M.alba_G0012659*), *bHLH* (*M.alba_G0009347*) and *bHLH3*

(*M.alba_G0016257*), shared the same expression pattern that was most highly expressed during the late stages of mulberry fruit development in ZJ (Figures 6 and S2–S4). Pearson correlation analysis showed that PCC (*r*) between the anthocyanin content and the expressions of the *MYB* (*M.alba_G0017209*), *MYB* (*M.alba_G0017689*), *bHLH* (*M.alba_G0012659*), *bHLH* (*M.alba_G0009347*) and *bHLH3* (*M.alba_G0016257*) in ZJ and ZZB at different post-flowering stages was 0.88, 0.86, 0.91, 0.78 and 0.68, respectively (Figures S5 and S6). KEGG analysis data showed significant enrichment of phytohormone signaling (Figure 3), in which the ethylene content was significantly accumulated, especially at later stages of ripening (Figure 1B). Further, comprehensive analysis of fruit color, anthocyanin contents and gene expression patterns at different developmental stages between both ZJ and ZZB varieties indicated that *ERF* (*M.alba_G0016603*), which is related to the ethylene responses, was involved in anthocyanin accumulation (Figures 6 and S7).

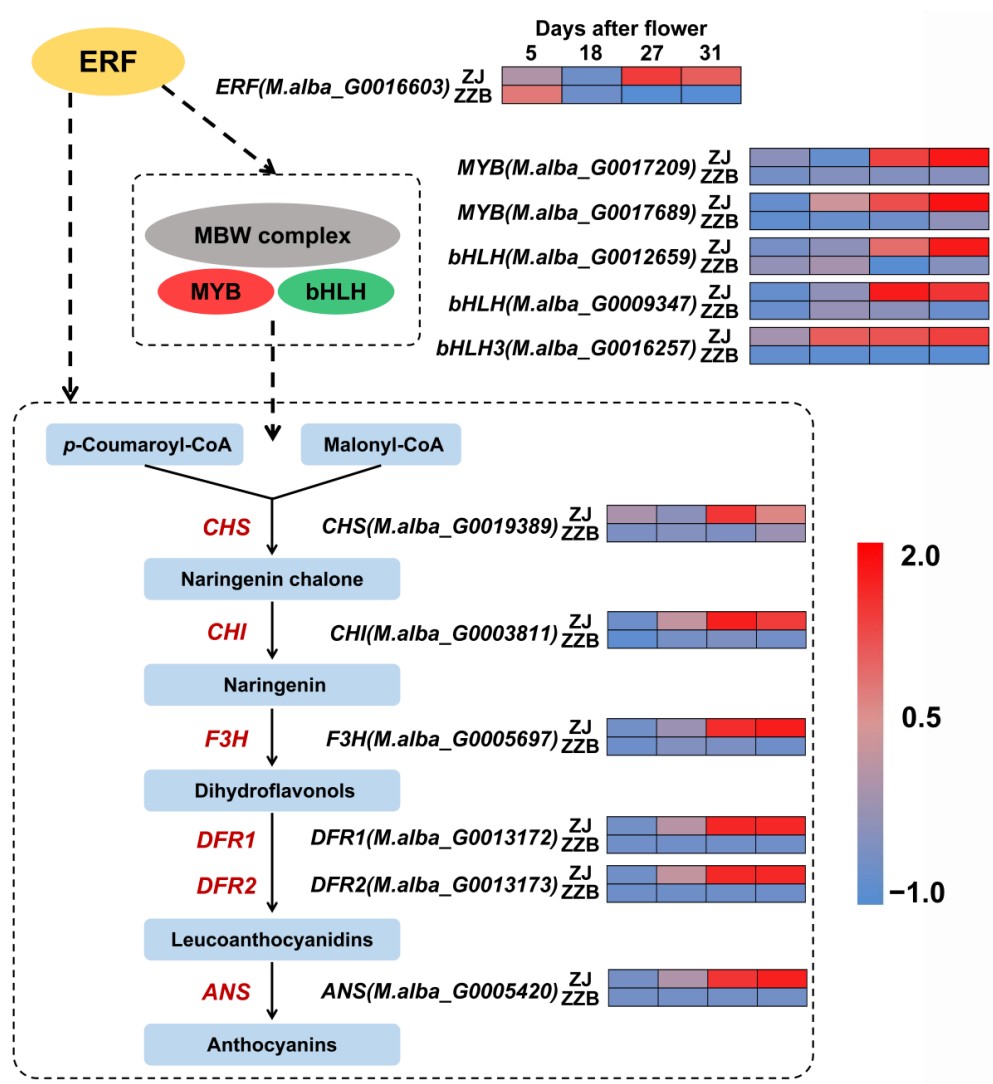

**Figure 6.** A simplified mechanistic model of the anthocyanin biosynthesis pathway and transcription of key genes in ZZB and ZJ. Each colored cell represents the average log2 (RPKM) value according to the color scale. Red represents high expression and blue represents low expression.

### 3.5. Ethylene-Induced Anthocyanin Accumulation in ZJ Cultivar Fruit

The mulberry fruits are climacteric. As shown in Figure 1A, with a large accumulation of ethylene, the *anthocyanin* accumulation rate and content of ZJ cultivar fruits increased significantly. To confirm that ethylene promotes fruit coloring and anthocyanin accumulation in mulberry, the ZJ cultivar fruits at 10 DAF in vivo and in vitro were sampled for

ethylene treatment with 0, 100, 300 and 500 mg/L, respectively (Figure 7A,B). The results demonstrated that the concentration of treated ethylene was positively correlated with an increase in anthocyanins in mulberry fruits (Figure 7C,D). However, high-concentration ethylene treatment can cause rapid senescence and shedding for mulberry fruit (data not shown).

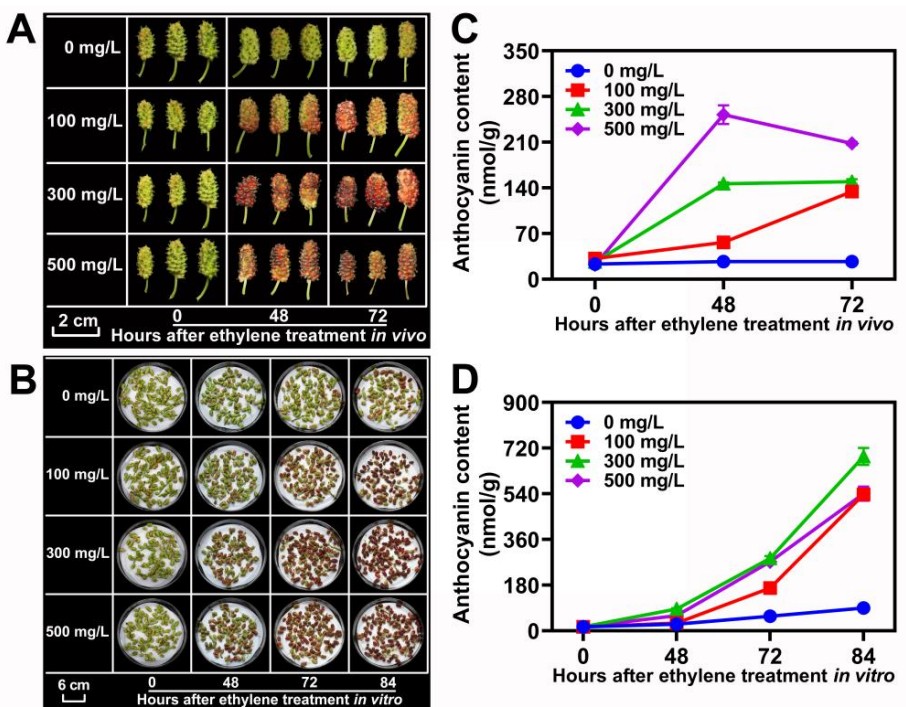

**Figure 7.** Ethylene increased the coloration and anthocyanin accumulation in ZJ fruits. (**A**,**B**) Morphology of ZJ fruits treated by specific concentrations of ethylene (0–500 mg/L) at different times in vitro and in vivo. (**C**,**D**) Anthocyanin content in ZJ fruits that were administered different concentrations of ethylene in vivo and in vitro. Data are mean± standard error, with three biological replicates.

Additionally, qRT-PCR results suggested that the expression levels of structural genes involved in anthocyanin biosynthesis (*CHS*, *CHI*, *F3H*, *DFR1*, *DFR2* and *ANS*) significantly increased as the ethylene concentration increased in ZJ fruits in vitro (Figure 8). Meanwhile, the TF genes, such as *MYB* (*M.alba_G0017209*), *bHLH* (*M.alba_G0012659*), *bHLH3* (*M.alba_G0016257*) and *ERF* (*M.alba_G0016603*), were significantly up-regulated by ethylene treatments (Figure 8). In particular, the expression of *ERF* (*M.alba_G0016603*) increased seven-fold at 72 h after ethylene treatment with 300 mg/L concentration (Figure 8). These results indicated that *ERF* (*M.alba_G0016603*) can specifically up-regulate expression in response to ethylene and promote anthocyanin accumulation in mulberry fruits.

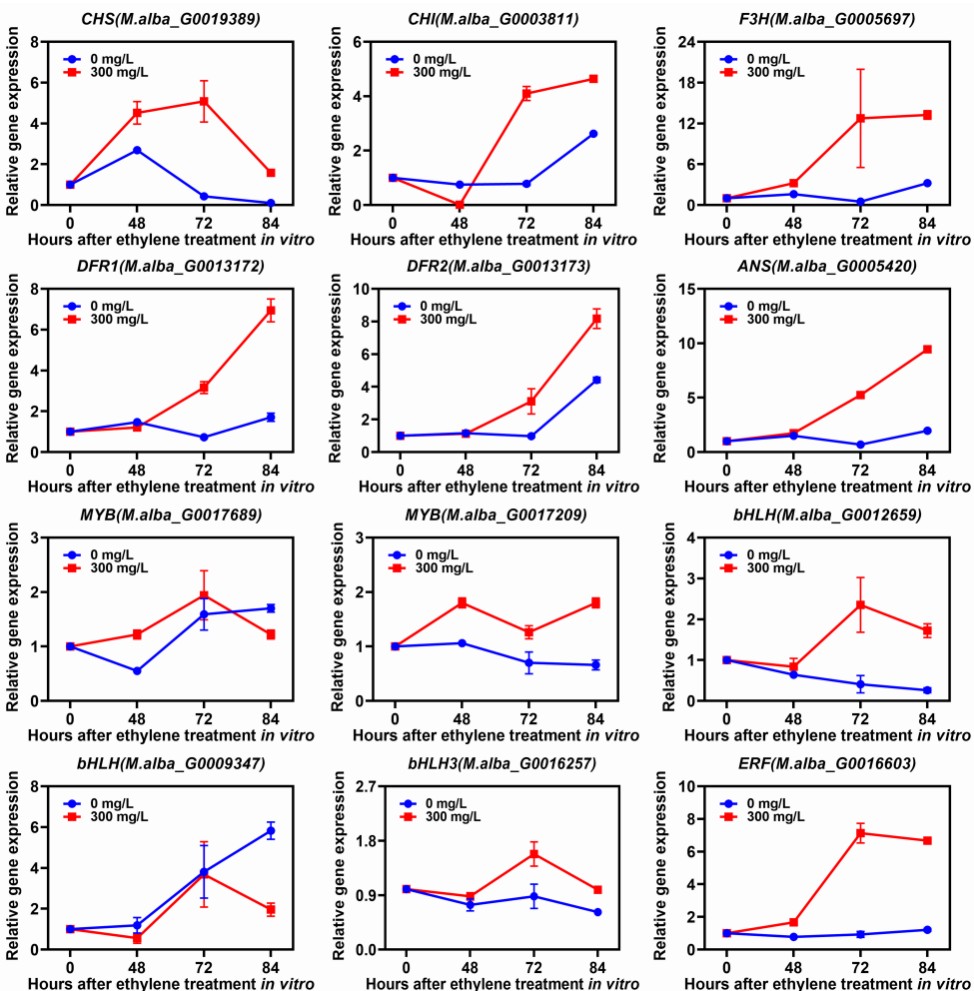

**Figure 8.** Relative expressions of different genes under in vitro treatment with 300 mg/L ethylene for a different time. Data represent the mean (n = 3) $\pm$ SE.

## 4. Discussion

As an advanced NGS technology, transcriptome sequencing is widely used to predict new genes, novel functions of old genes and genome evolution in plants. Comparing cultivars with different flesh colors can help identify different anthocyanin biosynthesis genes [8,31,32]. Comparative transcriptome sequencing of ZJ and ZZB revealed different expressions of genes regulating anthocyanin biosynthesis. In transcriptomic analysis, the expression abundances of *CHS* (*M.alba_G0019389*), *CHI* (*M.alba_G0003811*), *F3H* (*M.alba_G0005697*), *DFR1* (*M.alba_G0013172*), *DFR2* (*M.alba_G0013173*) and *ANS* (*M.alba_G0005420*) were significantly up-regulated and were strongly correlated with the anthocyanin content during fruit development and maturation (Figures 1C and 6). Similarly, in *Morus*, high transcript levels of these genes were detected in purple fruits of mulberry cultivars 'Da10' and 'Hongguo2', but not in white fruits of mulberry cultivars 'Zhenzhubai' and 'Baiyuwang', respectively [11,13].

MBW is the best-known regulatory complex in phycocyanin biosynthesis [12,33]. MYB and bHLH are the two main TFs regulating the expression of genes involved in anthocyanin biosynthesis [34,35]. Therefore, we also identified two DEGs from the MYB family and three DEGs from the bHLH family when comparing ZJ and ZZB (Figure 6). In detail, *MYB* (*M.alba_G0017209*), *MYB* (*M.alba_G0017689*), *bHLH* (*M.alba_G0012659*), *bHLH* (*M.alba_G0009347*) and *bHLH3* (*M.alba_G0016257*) were strongly expressed in ZJ ripening fruit, but their expressions in ZZB fruits were almost undetectable (Figure 6). Furthermore, PCC (*r*) between the anthocyanin content and the expressions of five TFs

in ZJ and ZZB at different DAF was 0.88, 0.86, 0.91, 0.78 and 0.69, respectively (Figure S5 and S6), indicating that they positively regulate anthocyanin biosynthetic genes. In addition, the combination of *bHLH3* (*M.alba_G0016257*) (PCC of 0.69) and MYBA activated the expression of anthocyanin biosynthetic genes, including *CYP75B1*, *ANS* and *UFGT*, and improved anthocyanin accumulation in mulberries [13].

Fleshy fruits are physiologically classified as climacteric or non-climacteric, based on their respiration and ethylene production at the onset of ripening [36]. The mulberry in this study is a typical climacteric fruit [26], which exhibits a burst of respiration and biosynthesis of the gaseous hormone ethylene at the onset of ripening (27 DAF) in ZJ and ZZB (Figure 1B). Ethylene plays an important role in controlling various aspects of color change, fruit softening and flavor formation during ripening in climacteric fruits [15]. A significantly increased anthocyanin content was observed in ZJ cultivar fruits (at 31 DAF) after the ethylene respiration peak. (Figure 1B,C). Ethylene treatment can significantly promote fruit coloring and anthocyanin accumulation of ZJ cultivar in vitro and in vivo (Figure 7), which is consistent with the findings of our previous study [28]. Meanwhile, the qRT-PCR analysis indicated that the abovementioned structural genes and TFs involved in anthocyanin biosynthesis were observably up-regulated under ethylene treatments at a concentration of 300 mg/L compared to 0 mg/L, such as *CHS* (*M.alba_G0019389*), *CHI* (*M.alba_G0003811*), *F3H* (*M.alba_G0005697*), *DFR1* (*M.alba_G0013172*), *DFR2* (*M.alba_G0013173*), *ANS* (*M.alba_G0005420*), *MYB* (*M.alba_G0017209*), *bHLH* (*M.alba_G0012659*) and *bHLH3* (*M.alba_G0016257*) (Figure 8).

ERFs are the final component in the ethylene signaling pathway, which have been confirmed to regulate color changes in fleshy fruits, such as apples [16,21] and pears [22,23]. In addition, *MaERF5* regulates anthocyanin biosynthesis in mulberry fruits by interacting with *MYBA* and *F3H* genes [28]. Comprehensive analysis of fruit color, anthocyanin contents and gene expression patterns at different developmental stages between both ZJ and ZZB varieties indicate that *ERF* gene, *ERF* (*M.alba_G0016603*), is a candidate key gene controlling the anthocyanins accumulation in mulberry fruits (Figure 6). The qRT-PCR analysis confirmed that *ERF* (*M.alba_G0016603*) was found to be strongly expressed following ethylene treatment at 300 mg/L (Figure 8). These results indicated that *ERF* (*M.alba_G0016603*) could be specifically up-regulated in response to ethylene and promote anthocyanin accumulation in mulberry fruits.

## 5. Conclusions

In summary, this study profiled the transcriptional changes in DEGs between the two mulberry genotype fruits (ZJ and ZZB) at different post-flowering stages. We successfully identified certain key structural genes (*CHS*, *CHI*, *F3H*, *DFR* and *ANS*) involved in anthocyanin biosynthesis in mulberry fruits. The ethylene-promoted fruit coloration and anthocyanin enhancement were confirmed in the ZJ cultivar and the markedly high-level expression of genes above was observed under ethylene treatment at 300 mg/L, rather than 0 mg/L. Overall, this work improves our knowledge about the process of anthocyanin biosynthesis and provides crucial information for the future exploration of the underlying molecular mechanism for anthocyanins regulating fruit color in *Morus*.

**Supplementary Materials:** The following supporting information can be downloaded at: https://www.mdpi.com/article/10.3390/horticulturae8100920/s1, Table S1: Primers used for qRT-PCR analysis in this study; Table S2: Statistics of the RNA-seq profiles between both genotypes (ZJ and ZZB) at seven different developmental stages; Figure S1: Heatmap for DEG structural genes involved in flavonoid–anthocyanin biosynthesis between ZZB and ZJ at different development stages; Figure S2: Heatmap for the DEGs of the *MYB* gene family between ZZB and ZJ at different development stages; Figure S3: Heatmap for the DEGs of the *bHLH* gene family between ZZB and ZJ at different development stages; Figure S4: Heatmap for the DEGs of *ERF* gene family between ZZB and ZJ at different development stages; Figure S5: Pearson correlation coefficient, PCC (*r*) between the anthocyanin content and the expressions of the *MYB* gene family in ZJ and ZZB at different development stages; Figure S6: Pearson correlation coefficient, PCC (*r*) between the anthocyanin content and the expressions of the *bHLH* gene family in ZJ and ZZB at different development

stages; Figure S7: Pearson correlation coefficient, PCC (*r*) between the anthocyanin content and the expressions of the *ERF* gene family in ZJ and ZZB at different development stages.

**Author Contributions:** Conceptualization, R.M., N.Z. and C.Y.; methodology, N.Z. and J.L.; software, Q.J. and Z.Z.; validation, R.M. and N.Z.; formal analysis, Z.D., Y.L. and C.Z.; investigation, R.M. and Y.L.; resources, C.Z.; data curation, R.M. and N.Z.; writing—original draft preparation, R.M.; writing—review and editing, R.M., N.Z. and C.Y.; visualization, Q.J. and J.L.; supervision, C.Z. and C.Y.; project administration, R.M. and C.Y.; funding acquisition, R.M. and C.Y. All authors have read and agreed to the published version of the manuscript.

**Funding:** This research was funded by the Key R&D Program of Hubei Province (2022BBA0065), the national key R&D program of China (2019YFD1000600), Key projects of the Natural Science Foundation of Hubei province, NSFH (2020CFA061) and the China Agriculture Research System of MOF and MARA.

**Data Availability Statement:** All data are available in the manuscript or the Supplementary Materials.

**Acknowledgments:** The authors thank Yinghua Tan (from Personal Biotechnology Co., Ltd., Shanghai, China) and Shaofang He (from Wuhan Carboncode Biotechnologies Co., Ltd.) for their technical support.

**Conflicts of Interest:** The authors declare no conflict of interest.

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
