# Peer review of "Transcriptomic Analysis Provides Insights into Anthocyanin Accumulation in Mulberry Fruits"

_horticulturae, doi:10.3390/horticulturae8100920_

Round 1
Reviewer 1 Report
I think authors set up the ethylene enhances anthocyanin accumulation in mulberry fruits and tried to some gene regulatory networks by using ethylene treatment. I have only minor comments to improve the quality of the paper.
Fig. 1, authors need to change the order- Zijing at the top and Zhenzhubai at the bottom since the graphs on the right shows this order so it is easy to compare.
Fig. 2A, authors need to change the color of the numbers inside, white is better than black. I can barely see 270 at the center.
Fig. 2C, authors need to change the color of the letters and size inside, white is better than black. I cannot see anything.
Author Response
I think authors set up the ethylene enhances anthocyanin accumulation in mulberry fruits and tried to some gene regulatory networks by using ethylene treatment. I have only minor comments to improve the quality of the paper.
Response: We sincerely thank the reviewer for the supportive comments. Many thanks!
Fig. 1, authors need to change the order- Zijing at the top and Zhenzhubai at the bottom since the graphs on the right shows this order so it is easy to compare.
Response: Thanks for your careful reviewing and advice. We have changed the order- Zijing at the top and Zhenzhubai at the bottom.
Please see Figure 1
Fig. 2A, authors need to change the color of the numbers inside, white is better than black. I can barely see 270 at the center.
Response: Thanks for your careful reviewing and advice. We have change the color of the numbers inside in Figure 2A.
Fig. 2C, authors need to change the color of the letters and size inside, white is better than black. I cannot see anything.
Response: Thanks for your careful reviewing and advice. We have change the color of the numbers inside in Figure 4A (Figure 2C). During the revision of the article, we changed Figure 2C to Figure 4A.
Please see Figure 4A
Reviewer 2 Report
Mo etal profiled transcriptomes associated with anthocyanin accumulation in mulberry fruits . The study has signifcance to advance our understanding the genes/pathways involved in anthocyanin compsoition and accumulation. However, the study needs to improve by including appropriate data analysis.
comments:
The study has two factors ( time -developmental stage and cultivars). However, the authors focused on DEGs between cultivars difference at different stage. I recommend to evaluate DEGs between the different stage of the same cultivar, and sebsequently the cluster analys should incude time or developmental stage as factor. I know the samples size (N=24) are relatively low, but it is still possible to run correlation network analysis as well.
Author Response
Mo etal profiled transcriptomes associated with anthocyanin accumulation in mulberry fruits. The study has signifcance to advance our understanding the genes/pathways involved in anthocyanin compsoition and accumulation. However, the study needs to improve by including appropriate data analysis.
Response: We sincerely thank the reviewer for the supportive comments. Many thanks!
comments:
The study has two factors (time -developmental stage and cultivars). However, the authors focused on DEGs between cultivars difference at different stage. I recommend to evaluate DEGs between the different stage of the same cultivar, and sebsequently the cluster analys should incude time or developmental stage as factor. I know the samples size (N=24) are relatively low, but it is still possible to run correlation network analysis as well.
Response: Thanks for your careful reviewing and advice. We have evaluate DEGs between the different stage of the same cultivar, and sebsequently the cluster analys should incude time or developmental stage as factor. Moreover, we run correlation network analysis as well.
Please see Line167-210, Figure 2 and Figure 3
We would like to take this opportunity to offer our gratitude to you for the helpful input about how to improve the scientific rigour and utility of our study and manuscript. Many thanks.
Round 2
Reviewer 2 Report
The authors addressed my comments.